# Identifying potential natural inhibitors of *Brucella melitensis* Methionyl-tRNA synthetase through an in-silico approach

**Adekunle Babajide Rowaiye**[1], **Akwoba Joseph Ogugua**[2]*, **Gordon Ibeanu**[3],
**Doofan Bur**[1], **Mercy Titilayo Asala**[1], **Osaretin Benjamin Ogbeide**[4], **Emmanuella Oshiorenimeh Abraham**[5], **Hamzah Bundu Usman**[6]

1 Department of Medical Biotechnology, National Biotechnology Development Agency, Abuja, Nigeria,
2 Department of Veterinary Public Health and Preventive Medicine, University of Nigeria, Nsukka, Nigeria,
3 Department of Pharmaceutical Science, North Carolina Central University, Durham, North Carolina, United States of America, 4 Department of Chemistry, University of Benin, Benin, Nigeria, 5 Department of Biochemistry, Caritas University, Enugu, Nigeria, 6 Department of Plant Science and Biotechnology, Federal University Gusau, Gusau, Nigeria

☉ These authors contributed equally to this work.
* ogugua.akwoba@unn.edu.ng

**Data Availability Statement:** All relevant data are within the manuscript and its Supporting Information files.

## Abstract

### Background

Brucellosis is an infectious disease caused by bacteria of the genus *Brucella*. Although it is the most common zoonosis worldwide, there are increasing reports of drug resistance and cases of relapse after long term treatment with the existing drugs of choice. This study therefore aims at identifying possible natural inhibitors of *Brucella melitensis* Methionyl-tRNA synthetase through an *in-silico* approach.

### Methods

Using PyRx 0.8 virtual screening software, the target was docked against a library of natural compounds obtained from edible African plants. The compound, 2-({3-[(3,5-dichlorobenzyl) amino] propyl} amino) quinolin-4(1H)-one (OOU) which is a co-crystallized ligand with the target was used as the reference compound. Screening of the molecular descriptors of the compounds for bioavailability, pharmacokinetic properties, and bioactivity was performed using the SWISSADME, pkCSM, and Molinspiration web servers respectively. The Fpocket and PLIP webservers were used to perform the analyses of the binding pockets and the protein ligand interactions. Analysis of the time-resolved trajectories of the Apo and Holo forms of the target was performed using the Galaxy and MDWeb servers.

### Results

The lead compounds, Strophanthidin and Isopteropodin are present in *Corchorus olitorius* and *Uncaria tomentosa* (Cat's-claw) plants respectively. Isopteropodin had a binding affinity score of -8.9 kcal / ml with the target and had 17 anti-correlating residues in Pocket 1 after molecular dynamics simulation. The complex formed by Isopteropodin and the target had a

**Funding:** The authors received no specific funding for this work.

**Competing interests:** The authors have declared that no competing interests exist.

total RMSD of 4.408 and a total RMSF of 9.8067. However, Strophanthidin formed 3 hydrogen bonds with the target at ILE21, GLY262 and LEU294, and induced a total RMSF of 5.4541 at Pocket 1.

## Conclusion

Overall, Isopteropodin and Strophanthidin were found to be better drug candidates than OOU and they showed potentials to inhibit the *Brucella melitensis* Methionyl-tRNA synthetase at Pocket 1, hence abilities to treat brucellosis. In-vivo and in-vitro investigations are needed to further evaluate the efficacy and toxicity of the lead compounds.

## Author summary

The cure for brucellosis involves a long course of treatment with a combination of antibiotics. However, some of the drugs are not recommended for very young children and pregnant women. Moreover, cases of relapse and resistance to these drugs are reported. With the Brucella Methionyl-tRNA synthetase as a target, molecular docking and virtual screening was used to identify possible drug candidates from a library of 1524 compounds obtained from edible African plants. Two lead compounds, Strophanthidin and Isopteropodin usually present in *Corchorus olitorius* and *Uncaria tomentosa* (Cat's claw) plants showed potentials to inhibit the *Brucella melitensis* Methionyl-tRNA synthetase. Their bioactivities were also confirmed in their molecular dynamic simulation with the target protein. Consequently, both compounds have potentials for safety and efficacy in the treatment of brucellosis.

## Introduction

Brucellosis is an infectious disease caused by bacteria of the genus *Brucella*. The species are Gram-negative intracellular coccobacilli that occur in a wide variety of animals including cattle, sheep, goats, pigs, other livestock as well as humans [1]. There are 12 species of *Brucella* based on specificity of host [2]. Although, *Brucella* species are often associated with certain hosts, they infect others apart from their preferred hosts. Being basically a disease of animals, most human brucellosis cases are traceable to infected animals or their products [3]. Hence, its control in human populations is targeted at the animals. Most infections in humans are due to contact with contaminated materials. The disease therefore has a major occupational disposition among livestock workers, veterinarians, abattoir workers, so also hides, skin and wool workers as well as laboratory personnel [4,5]. To the general public, brucellosis is mainly transmitted through the consumption of unpasteurized contaminated milk or its products [6,7]. In few occasions, human-to-human transmissions have been recorded through sexual contact, blood transfusion, bone marrow transplant, obstetrical manipulations during child birth and congenital means [4,8,9]. Brucellosis however is noted as the most common zoonosis worldwide with more than 500,000 cases recorded annually [10].

The disease is well controlled in most developed countries [11], but common in Africa, South America, Asia, the Caribbean, Middle East and the Mediterranean basin [2,12,13]. In livestock production, the major economic effects are due to abortion, premature birth, reduced milk production, repeat breeding and cost of veterinary care [14]. In humans, the disease

results in loss of manpower as well as huge costs in medical care [15]. Thus, the control of the disease in most developed countries has resulted in significant economic gains as well as reduction in human cases. However, in developing countries the disease is still of major economic and public health importance. This is mainly due to lack of well-defined control policies as well as the lifestyle of high-risk persons who are mostly uninformed about the disease [16]. In controlling brucellosis, many countries embark on or consider the actions compatible with their tradition and resources. Methods of controlling brucellosis therefore are hinged on diagnosis, control, increasing the awareness of the disease and vaccination [17]. Brucellosis remains a largely neglected disease especially in developing countries [3]. In sub-Sahara Africa, there has been little attention paid towards the control and prevention of brucellosis except in South Africa [18]. The control of brucellosis in Africa is hindered by many factors. The farming system is basically traditional. Nomadism which accounts for as high as 95% of cattle production in many West African countries [3] involves uncontrolled movement of livestock: a major risk factor in the spread of brucellosis [16]. Brucellosis is therefore noted to impact negatively on human and animal health, hampers social and economic progress as well as food security in developing economies [19].

*Brucella* remains a potential bio-terroristic agent and moreover, treatment of the disease is quite difficult in affected people because of the ability of the organism to evade the host immune system and reside in the cell for extended periods [20]. Most drugs currently used to treat *Brucella* infection have not been relatively effective. This is because *Brucella* activates the cAMP/protein kinase A pathway which is crucial for the survival and establishment of *Brucella* within macrophages. Inside the cells, they inhibit programmed cell death leading to long survival in the cells. Effective antimicrobial treatment for sufficient length of time with drugs including, doxycycline in combination with rifampin or streptomycin [21] and other recommended drugs for the treatment of brucellosis [22], have been hampered by relapses and therapeutic failures [8,23]. Also, prevalence of drug resistance genes is being reported in *Brucella* species [24,25]. Resistance to these drugs of choice have been observed in Turkey [22], China [26], Brazil [27], Kazakhstan [28], Norway [29] and Egypt [30] Such reports of antibiotic resistance are rendering the use of antibiotics almost useless in treatment of brucellosis [31]. In the same vein, doxycycline the most effective of these drugs, is contraindicated in pregnant women and children below eight years of age [32,33] This underscores the need to search for alternatives to the current long term chemotherapy of brucellosis with these drugs. Such new agents need to be able to penetrate and function within the macrophage cytoplasm, inexpensive, non-toxic and more effective than the drugs traditionally used to treat the disease.

Plants have long been viewed as a common source of remedies, either in the form of traditional preparations or as pure active principles. Many antibacterial compounds that may prove to be useful leads for antibacterial drug discovery have been derived from medicinal plants [34]. These plants have had a great influence on the daily lives of people living in developing countries, as the population in these countries cannot generally afford the cost of Western medicines. Hence, natural products of plant biodiversity have received considerable attention as potential antibacterial agents since they are a proven template for the development of new antimicrobials [35]. Natural compounds have been utilized and/or chemically modified by humans to prevent, treat and cure diseases since 5000 BC and the WHO intends to integrate traditional medicine into National Health Systems (NHS) globally [36]. This provides an opportunity for building safe, affordable and effective NHS especially for Third world countries, rich in both medicinal plant resources and traditional medicine knowledge. These plants could be relied on as sources of agents that would act on well-defined molecular bacterial targets, to improve the therapeutic effects lacking in the traditional antimicrobials.

Availability of sequenced genome of *Brucella* species has offered new options in the search for drugs targeting enzymes that could be of use due to pathogen-host physiological and biochemical differences. The methionyl-tRNA synthetase, which is a member of the aminoacyl tRNA synthetase group, has been identified as being very important for its roles in protein synthesis due to its recognition of initiator tRNA and tRNA delivering methionine for protein chain elongation [37]. According to Ojo *et al.* [38], methionyl-tRNA synthetase is promising as a good target for brucellosis drug development. Therefore, lead compounds targeting the enzyme could be useful and offer good alternatives for the treatment of brucellosis. This study hypothesizes that compounds targeting the enzyme can be found in edible African herbs. The aim of this study is to use in-silico method to identify compounds of plant origin that can inhibit the activity of *Brucella melitensis* methionyl-tRNA synthetase and serve as remedies for brucellosis. This will pave the way for subsequent studies testing for the effectiveness of the identified compounds (*in-vitro* and *in-vivo*) against *B. melitensis*.

## Method

The *SWISS-MODEL* homology modeling server was used to model the target protein after the crystal structure of methionyl-tRNA synthetase MetRS from *Brucella melitensis* (PDB ID: 5K0S.1.A) [39,40]. The structure of the target protein was visualized using PyMOL [41], analyzed using the VADAR 1.8 server [42] and validated using the MolProbity server [43].

A library of 1,524 phytoconstituents belonging to different classes of secondary metabolites was collected from the results of the phytochemical analyses of edible African plants found in literature. The Structure Data File (sdf) formats of the 3D chemical structures of these compounds were downloaded from the PubChem database [44]. All ligands were loaded on the PyRx 0.8 software, their geometries were minimized and they were converted into the pdbqt format in readiness for molecular docking [45].

Docking of all the ligands against the target was performed using the AutoDock Vina tool of PyRx 0.8 software. The grid parameters were set at Center—x: 26.4524, y: 19.1969, z: 22.6112 and Dimensions–x: 64.4178, y: 72.5182, z: 84.2900. The setting for the docking was the universal force field (*UFF*) and conjugate gradient algorithm [45]. The 2-({3-[(3,5-dichlorobenzyl) amino] propyl} amino) quinoline-4(1H)-one (OOU) (PubChem ID 18353708) which is the co-crystallized ligand of the target protein was used as the reference compound. From the docking results, all docking scores higher than the binding affinity score of OOU (reference compound) with the target were screened out.

The predictions for molar refractivity, saturation and promiscuity for the front runner compounds were obtained from the SwissADME server and screening was performed based on established medicinal chemistry criteria [46]. Screening for absorption, distribution, metabolism, elimination, and toxicity (ADMET) properties was performed using the pkCSM server [47]. Further screening of the front runner compounds for bioactivity was performed with the Molinspiration server [48]. The PLIP webserver was used to decipher the hydrogen bonds, halogen bonds and hydrophobic interactions between residues of the target and the lead compounds [49].

A molecular dynamic simulation study of the apo and holo forms of the target protein was performed using the Galaxy and MDWeb servers [50,51]. Analyses of the time-resolved trajectory were done using parameters such as root-mean-square deviation (RMSD), root mean square fluctuation (RMSF), radius of gyration (RoG), B- factor, principal component analysis (PCA), and dynamical cross-correlation matrix (DCCM) [50,51]. The $LD_{50}$ of the lead compounds are to be determined at the *in-vivo* validation of the results of this research.

The FASTA format of the amino acid sequences of the target protein (P59078) was obtained from the UniProtKB database [52]. Sequences were placed on the BLAST tool of the NCBI server and the settings were, PDB protein for database, Homo sapiens (Taxid 96906) for organism, and blastp for algorithm [53].

## Results

### Analysis of the structure of the target

The modelled target protein had 507 residues with a 100% similarity identity with *Brucella melitensis* methionyl-tRNA synthetase (BrMelMetRS) (PDB: 5K0S) and also a qualitative model energy analysis (QMEAN) value of 0.76 and global model quality estimate (GMQE) value of 0.96. Resolved by X-ray diffraction method, the crystal structure of BrMelMetRS (PDB: 5K0S) showed a resolution of 2.45 Å and R-Value Free of 0.256 (Fig 1). The secondary structures of the target included 49% alpha helix, 22% beta sheets and 28% coils. The total solvent-accessible surface area (SASA) was 22269.0 (Å)$^2$. Ramanchandran analysis revealed that in terms of geometry, the target protein had 1.21% poor rotamers, 97.1% favoured rotamers of which 99.6% were in allowed regions, 0.40% Ramachandran outliers, 98.02% ramanchandran favoured, 0.00% Cβ deviations (>0.25Å), and Rama distribution Z-score of 1.11 ± 0.36, 0.07% bad bonds and 0.48% bad angles (Fig 2). With regards to low-resolution criteria, there were 0.8% carbon-alpha based low-resolution annotation method (CaBLAM) outliers and 0.60% carbon-alpha geometry outliers.

### Drug-likeness properties and other molecular descriptors of ligands

For the reference and lead compounds, the drug-likeness properties such as hydrogen bond acceptor (HBA), hydrogen bond donor (HBD), log P, molecular weight, and topological surface area (TPSA) did not exceed 10, 5, 500 g/mol and 140 Å respectively (Fig 3 and Table 1). Furthermore, the reference and lead compounds' molar refractivity ranged from 40 to 130,

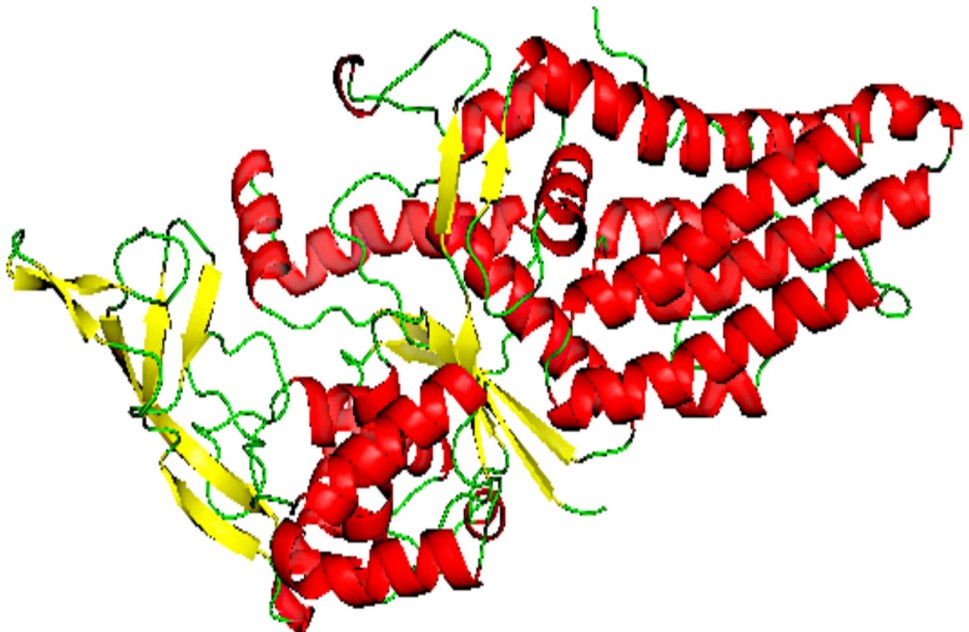

**Fig 1. The cartoon structure of modeled BrMelMetRS.** Beta sheets in yellow, alpha helix in red, and loops in green.

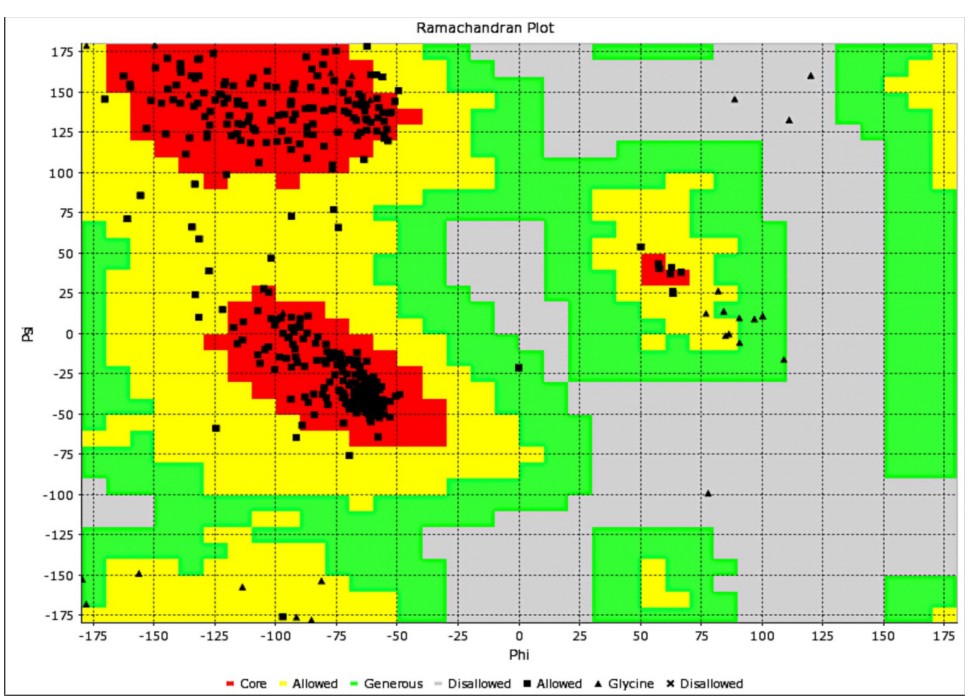

**Fig 2. Ramachandran plot of modeled BrMelMetRS.**

despite the fact that their number of rotatable bonds did not surpass 10. In terms of bioactivity, OOU and Strophanthidin had enzyme inhibition prediction values larger than 0.00, whereas Isopteropodin had a value less than 0.00. All the compounds had all their bioactivity prediction values greater than -5.00. From the bioavailability radars, all the compounds were within the range for drug-likeness properties of size, lipophilicity, solubility, polarity and flexibility (Fig 4). While OOU was slightly unsaturated, Strophanthidin and Isopteropodin were within the saturation range (above 0.25).

## The ADMET properties of ligands

From Table 2, the water solubility values for both the leads and reference compounds were greater than -6.0 log mol/L. The values for OOU and Isopteropodin's Caco-2 permeability (log Papp in 10–6 cm.s-1) were larger than 0.9, while Strophanthidin's value was less than 0.9. For all of the compounds, the human intestine absorption (percentage absorbed) values were greater than 30%. Similarly, all the compounds had skin permeability (LogKp) values less than -2.5 (Table 2). Remarkably, OOU was predicted to be inhibitor of both P-glycoprotein I and II, while the lead compounds were not. However, all compounds were P-glycoprotein substrates.

In terms of distribution, Strophanthidin had a CNS permeability (Log PS) value less than -3.0, while Isopteropodin and OOU had values larger than -3.0 but less than -2.0. The OOU and Isopteropodin had their volume of distribution steady state (Log VDss) values of more than 0.45, although Strophanthidin had a value of less than 0.15. All compounds had their BBB permeability (log BB) larger than -1.0 but less than 0.3. Similarly, all the compounds had their fraction unbound values greater than 0.1. With regards to metabolism, all compounds were non-inhibitors of cytochrome P450 2C19 and 2C9 enzymes and all substrates of cytochrome P450 3A4. Only OOU was an inhibitor of cytochrome P450 2D6, 1A2 and 3A4 enzymes and a substrate of cytochrome P450 2D6 (Table 2).

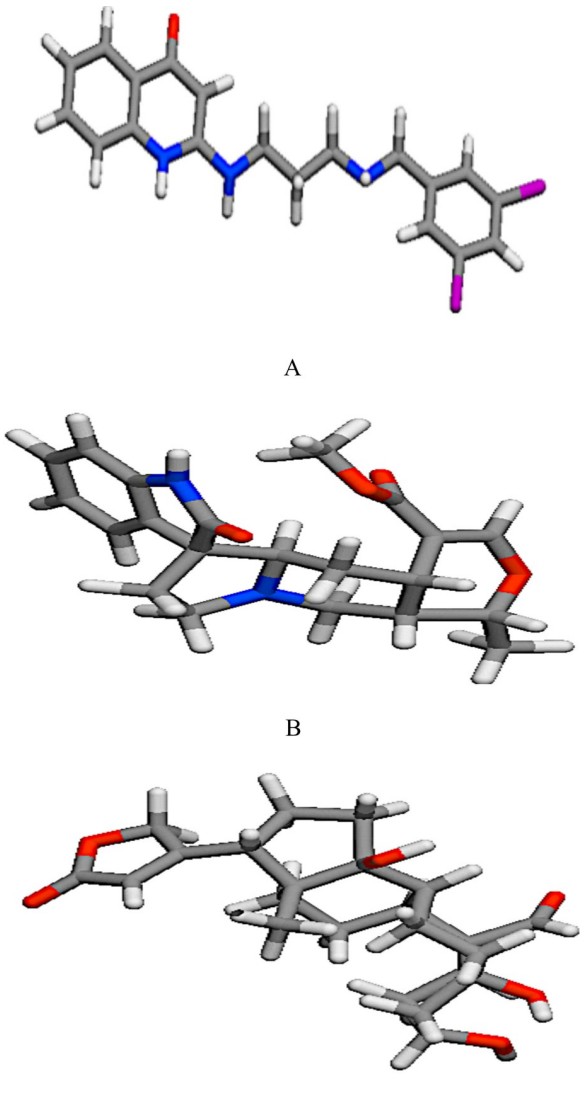

**Fig 3.** The stick model of the 3D structures of the reference and lead compounds (a) OOU (b) Isopteropodin (c) Strophanthidin.

In terms of excretion, Strophanthidin recorded the lowest total clearance (log ml/min/kg), whereas OOU showed the highest. Strophanthidin was not a substrate of renal OCT2, only OOU and Isopteropodin were. All the compounds showed no AMES toxicity, no dermotoxocity and were non-inhibitors of hERG I proteins. However, only OOU was predicted to be a blocker of hERG II and only strophanthidin was not hepatotoxic. The values for maximum tolerated dose (log mg/kg/day), oral rat acute toxicity (LD50) (mol/kg) and oral rat chronic toxicity (log mg/kg/day) were highest in OOU, Isopteropodin, and strophanthidin respectively. For lead compounds and the reference, the *T. Pyriformis* toxicity (log g/L) values were all larger than -0.5. Minnow toxicity (log mM) was less than 0.3 only for Isopteropodin (Table 2)

## Analysis of molecular docking scores

Isopteropodin had the lowest binding score with the target protein (Table 3).

**Table 1. Chemical and physical properties of reference and lead compounds.**

| Descriptors | OOU(reference) | Isopteropodin | Strophanthidin |
|---|---|---|---|
| Chemical formula | C19H19Cl2N3O | C21H24N2O4 | C23H32O6 |
| PubChem ID | 18353708 | 98363 | 6185 |
| Molecular Weight (g/mol) | 376.3 | 368.4 | 404.5 |
| XLogP3 | 5 | 1.6 | 0.6 |
| HBD count | 3 | 1 | 3 |
| HBA count | 4 | 5 | 6 |
| Rotatable bond count | 7 | 2 | 2 |
| TPSA (Å$^2$) | 56.92 | 67.9 | 104 |
| PAIN Alerts | None | None | None |
| Molar Refractivity | 105.6 | 106.47 | 106.16 |
| G-Protein CR Ligand | 0.31 | 0.37 | 0.08 |
| Ion channel modulator | 0.05 | 0.25 | 0.07 |
| Protein Kinase Inhibitor | 0.27 | -0.34 | -0.46 |
| Nuclear Receptor ligand | -0.2 | 0.07 | 0.52 |
| Protease Inhibitor | 0.1 | -0.02 | 0.01 |
| Enzyme Inhibitor | 0.24 | -0.02 | 0.79 |

## Binding site analyses

The reference and two lead compounds bound at residues ILE 12, TYR 14, VAL 229, TRP 230, ALA 233, LEU 234, GLY 262, ILE 265, PHE 268, PHE 293, and LEU 294 and could all be found in Pocket 1 of the target (Figs 5 and 6, Tables 4 and S1 Fig). The BrMelMetRS–Strophanthidin complex formed the highest number of intermolecular hydrogen bonds with the target. In terms of bond angle, Isopteropodin and Strophanthidin each formed one bond less than 130° at ILE12 and LEU294 respectively. The OOU, Isopteropodin, and Strophanthidin formed one, one and two bonds respectively that were greater than 130°. With reference to the donor to acceptor distance, OOU made no hydrogen bond within the range of 2.5–3.2 Å, none within the range of 3.2–4.0 Å, and only one (TYR14A) above 4.0 Å with the target. Isopteropodin formed one hydrogen bond (at ILE12A) within the range of 2.5–3.2 Å, and one (GLY262) within the range of 3.2–4.0 Å. Strophanthidin formed one hydrogen bond (at LEU294A) within the range of 2.5–3.2 Å, and two (at ILE21 and GLY262) within the range of 3.2–4.0 Å (Table 4). From Table 5, the BrMelMetRS–OOU complex had the highest number (12) of hydrophobic interactions and it was the only one that had a halogen bond at ASP232.

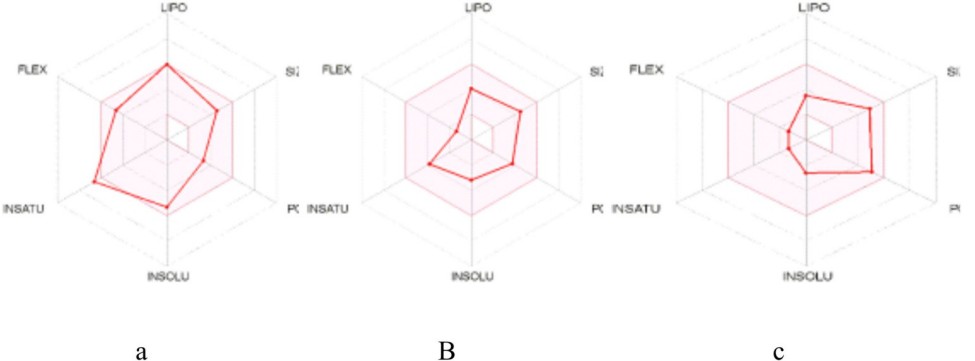

**Fig 4.** The bioavailability radars for reference and lead compounds (a) OOU (b) Isopteropodin (c) Strophanthidin.

**Table 2. ADMET properties of reference and lead compounds.**

| Variables | OOU (Reference) | Isopteropodin | Strophanthidin |
|---|---|---|---|
| **Absorption** | | | |
| Solubility in water (log mol/L) | -4.026 | -3.521 | -4.473 |
| Permeabilities of Caco-2 (log Papp in 10–6 cm.s-1) | 1.058 | 1.119 | 0.813 |
| Intestinal absorption in humans (% absorbed). | 89.488 | 96.483 | 73.206 |
| Permeability of skin (log Kp) | -2.739 | -3.767 | -3.909 |
| P-glycoprotein substrate (Yes/No) | Yes | Yes | Yes |
| P-glycoprotein I inhibitor (Yes/No) | Yes | No | No |
| P-glycoprotein II inhibitor (Yes/No) | Yes | No | No |
| **Distribution** | | | |
| Volume of Distr. Steady State (human) (log L/kg) | 1.347 | 0.845 | 0.143 |
| Fraction unbound (human) | 0.15 | 0.357 | 0.38 |
| Permeability of BBB (log BB) | 0.172 | 0.035 | -0.602 |
| Permeability of CNS (log PS) | -2.108 | -2.307 | -3.098 |
| **Metabolism** | | | |
| Substrate of cytochrome P450 2D6 (Yes/No) | Yes | No | No |
| Substrate of cytochrone P450 3A4 (Yes/No) | Yes | Yes | Yes |
| Inhibitor of cytochrome P450 1A2 (Yes/No) | Yes | No | No |
| Inhibitor of cytochrome P450 2C19 (Yes/No) | No | No | No |
| Inhibitor of cytochrome P450 2C9 (Yes/No) | No | No | No |
| Inhibitor of cytochrome P450 2D6 (Yes/No) | Yes | No | No |
| Inhibitor of cytochrome P450 3A4 (Yes/No) | Yes | No | No |
| **Excretion** | | | |
| Total Clearance (log ml/min/kg) | 0.951 | 0.886 | 0.624 |
| Substrate of Renal OCT2 (Yes/No) | Yes | Yes | No |
| **Toxicity** | | | |
| AMES toxicity (Yes/No) | No | No | No |
| Max. Tolerated dose (human) (log mg/kg/day) | -0.088 | -1.088 | -0.487 |
| Blocker of hERG I (Yes/No) | No | No | No |
| Blocker of hERG II (Yes/No) | Yes | No | No |
| Oral Rat Acute Toxicity (LD50) (mol/kg) | 2.187 | 2.763 | 2.357 |
| Oral Rat Chronic Toxicity (log mg/kg/day) | 1.4 | 1.771 | 1.833 |
| Liver toxicity (Yes/No) | Yes | Yes | No |
| Sensitization of skin (Yes/No) | No | No | No |
| Toxicity to T. Pyriformis (log µg/L) | 0.42 | 0.526 | 0.306 |
| Toxicity to Minnows (log mM) | 0.488 | -0.364 | 2.387 |

## Molecular dynamics simulation

Fig 7 reveals the structures of the apo and holo forms of BrMelMetRS after a 2-nanosecond molecular dynamics simulation. The BrMelMetRS- Isopteropodin and BrMelMetRS-OOU

**Table 3. Docking scores of ligands against the target.**

| Ligand | Binding Score (Kcal/mol) |
|---|---|
| OOU (reference) | -8.6 |
| Isopteropodin | -8.9 |
| Strophanthidin | -8.6 |

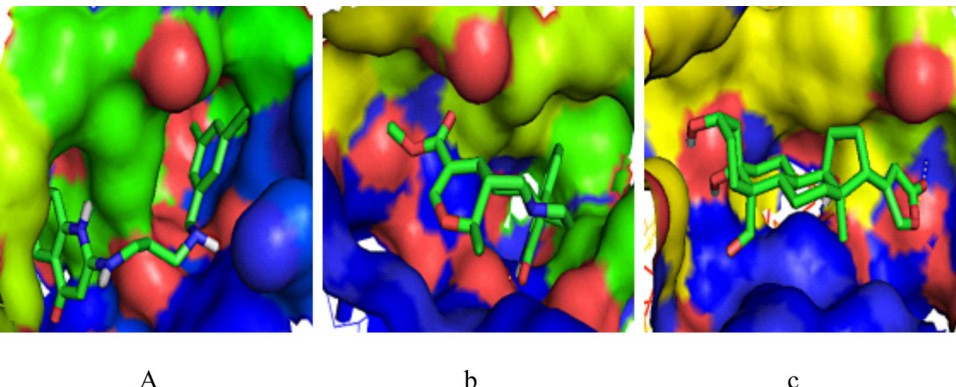

**Fig 5.** Binding site of the target showing interaction with reference and lead compounds. (a) BrMelMetRS-OOU complex, (b) BrMelMetRS-Isopteropodin complex (c) BrMelMetRS-Strophanthidin complex.

complexes had the same values of average and total RMSD which were higher than that of the BrMelMetRS- Strophanthidin complex. However, the BrMelMetRS-OOU trajectory peaked at time frame 15 (0.235) as compared with that of the BrMelMetRS- Isopteropodin complex which peaked at time frame 11 with a slightly lower RMSD value (0.231) (S1 Fig and Table 6). In terms of RMSD distribution, all the 20 peaks of the apo and holo forms of the target were found within 0.0–0.49 Å (S2 Fig and Table 6).

The BrMelMetRS-Isopteropodin complex exhibited the greatest average and total RMSF values of all the holo structures. The least was the BrMelMetRS- Strophanthidin complex. However, at the regional level (Pocket 1), the BrMelMetRS-Strophanthidin complex had the highest values for average and total RMSF while that of the BrMelMetRS- Isopteropodin complex was the lowest (S3 Fig and Table 6).

The cumulative of the first three highest principal components (PC1, PC2, and PC3) for all the holo forms of the target represented less than 50% of the total variance (S4 Fig and Table 6). The BrMelMetRS- Isopteropodin complex had the highest total and average global motions of all the holo forms. At the regional level (Pocket 1), the BrMelMetRS-OOU complex had the highest average and motions followed by the BrMelMetRS- Strophanthidin complex. Overall, the best conformations in terms of the greatest global motions were PC3, PC2, and PC2 for BrMelMetRS-OOU, BrMelMetRS-Isopteropodin complex, and

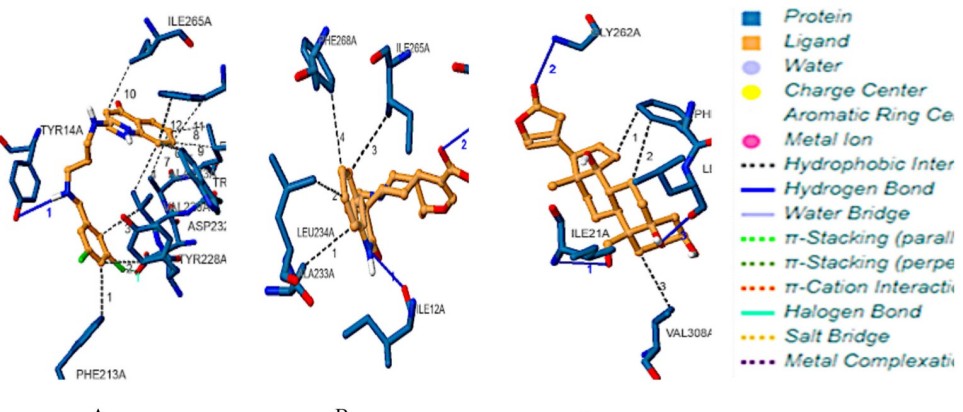

**Fig 6.** Interactions of target with reference and lead compounds (a) BrMelMetRS-OOU complex, (b) BrMelMetRS-Isopteropodin complex (c) BrMelMetRS-Strophanthidin complex.

**Table 4. Analysis of Hydrogen bond interactions between target and ligands.**

| Protein-ligand Complexes | No. of bonds | Residues | Distance (H-A) | Distance (D-A) | Bond angle |
|---|---|---|---|---|---|
| BrMelMetRS–OOU | 1 | TYR14A | 3.06 | 4.08 | 166.07 |
| BrMelMetRS—Isopteropodin | 2 | ILE12A | 2.15 | 2.84 | 123.04 |
| | | GLY262A | 3.14 | 3.85 | 130.68 |
| BrMelMetRS—Strophanthidin | 3 | ILE21A | 3.1 | 3.84 | 132.63 |
| | | GLY262A | 2.48 | 3.21 | 130.82 |
| | | LEU294A | 2.35 | 2.82 | 107.99 |

BrMelMetRS-Strophanthidin complexes respectively and the same for regional motions. The PCA cosine content of the dominant motions related to PC1 for all the holo forms of the target did not get to 1.0 (Table 6).

In terms of average radius of gyration along the trajectory, the BrMelMetRS-OOU complex had the highest value followed by the BrMelMetRS-Isopteropodin complex. However, the BrMelMetRS-Isopteropodin complex had the widest range of gyration (S5 Fig and Table 6). At the global and regional (Pocket 1) levels, B-factor values were highest in the BrMelMetRS-OOU complex (S6 Fig and Table 6). Additionally, the dynamic cross-correlation analysis revealed that of the 31 residues of the Pocket 1, the BrMelMetRS-OOU complex had the highest number of anti-correlating residues (S7 Fig and Table 6).

## BLAST

The closest structures to the BrMelMetRS in the human proteome proteins were three unnamed protein products CBX51367.1, CAE90564.1 and CAE89160.1 (Table 7). The CBX51367.1 had a query cover of 91% while the other two showed short alignments each with 22% query cover.

**Table 5. Hydrophobic interactions and Halogen bonds.**

| Protein-ligand Complexes | Hydrophobic Interaction | | Halogen Bonds | | | |
|---|---|---|---|---|---|---|
| | Residue | Distance | Residue | Distance | Donor angle | Acceptor angle |
| BrMelMetRS—OOU | PHE213A | 3.62 | ASP232A | 3.36 | 140.52 | 124.94 |
| | TYR228A | 3.69 | | | | |
| | VAL229A | 3.53 | | | | |
| | VAL229A | 3.69 | | | | |
| | TRP230A | 3.91 | | | | |
| | TRP230A | 3.59 | | | | |
| | ALA233A | 3.82 | | | | |
| | LEU234A | 3.34 | | | | |
| | LEU234A | 3.92 | | | | |
| | ILE265A | 3.78 | | | | |
| | PHE268A | 3.9 | | | | |
| | PHE268A | 3.58 | | | | |
| BrMelMetRS—Isopteropodin | ALA233A | 3.62 | | | | |
| | LEU234A | 3.99 | | | | |
| | ILE265A | 3.68 | | | | |
| | PHE268A | 3.68 | | | | |
| BrMelMetRS—Strophanthidin | PHE293A | 3.38 | | | | |
| | PHE293A | 3.91 | | | | |
| | VAl308A | 3.95 | | | | |

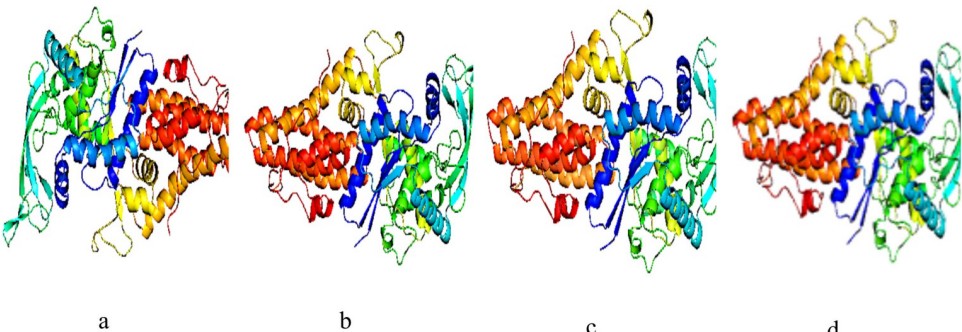

**Fig 7.** Cartoon model of the apo and holo forms of the target (after MDS). (a) BrMelMetRS (b) BrMelMetRS-OOU complex (c) BrMelMetRS-Isopteropodin complex (d) BrMelMetRS-Strophanthidin complex.

## Discussion

### The target

The structure of a protein determines its biological function [54]. The qualitative model energy analysis (QMEAN) is a composite scoring function assessing the major geometrical (global and local) aspects of protein structures and the assessment of the model ranges from 0 to 1 (with one being perfect) [55]. With a QMEAN value of 0.76, the modelled BrMelMetRS (PDB: 5K0S.1.A) has a high structural quality. Similarly, the global model quality estimation (GMQE) score evaluates the structural quality of models using evolutionary information and it is expressed as a value between 0 and 1 (with one being the most reliable) [56]. A value of 0.96 suggests a high reliability of the modelled target. Also, a ramanchandran favoured value greater than 98% and a Rama distribution Z-score less than 2 are suggestive of good stereo-chemistry of the modeled target [43].

### Drug-likeness and bioactivity

A compound's drug-likeness is determined by how similar it is to existing drugs in terms of structural and physicochemical properties [57]. In terms of drug-likeness, the reference and lead compounds do not violate the Ghose, Lipinski, and Veber rules seeing that the values of their HBA, HBD, log P, molecular weight, TPSA, molar refractivity, and number of rotatable bonds are within accepted range [58]. Therefore, they are all predicted to have good size, polarity and flexibility which positively correlate to good bioavailability. However, the bioavailability radar of OOU suggests that it is slightly unsaturated with fraction of carbons in the sp3 hybridization ($Fsp^3$) value less than 0.25 [59,60]. Both complexity (as assessed by $Fsp^3$), the presence of chiral centers, and saturation which is linked with solubility are all crucial in drug development [59]. Therefore, OOU would be a poor drug candidate.

The biological activity profiles of possible drug candidates must take into account human metabolism because drugs interact with several molecular targets in the body [61]. With respect to bioactivity, Strophanthidin is predicted to have the highest enzyme inhibition activity while Isopteropodin has the lowest (Table 1).

### ADMET

The ADMET properties of candidate compounds are the main reason of high attrition rates in drug discovery [62]. Aqueous solubility is a critical physicochemical feature that influences pharmacokinetic properties and drug formulations [63]. From Table 2, Isopteropodin is the

**Table 6. MDS of the apo and holo forms of the target (a summary).**

| MDS Parameters | BrMelMetRS-Apo | BrMelMetRS—OOU | BrMelMetRS—Isopteropodin | BrMelMetRS—Strophanthidin |
|---|---|---|---|---|
| **RMSD** | | | | |
| Total RMSD | 4.437 | 4.408 | 4.408 | 4.356 |
| Average RMSD | 0.2112 | 0.2099 | 0.2099 | 0.2074 |
| Highest RMSD | 0.234 | 0.235 | 0.231 | 0.23 |
| Lowest RMSD | 0 | 0 | 0 | 0 |
| Time Frame of Highest RMSD | 16 | 15 | 11 | 3 |
| Time Frame of Lowest RMSD | 1 | 1 | 1 | 1 |
| RMSD Peak Distribution | | | | |
| 0.00–0.49A | 20 | 20 | 20 | 20 |
| 0.50–0.99A | 0 | 0 | 0 | 0 |
| 1.00–1.49A | 0 | 0 | 0 | 0 |
| 1.50–1.99A | 0 | 0 | 0 | 0 |
| 2.00–2.49A | 0 | 0 | 0 | 0 |
| 2.50–2.99A | 0 | 0 | 0 | 0 |
| 3.00–3.49A | 0 | 0 | 0 | 0 |
| **RMSF** | | | | |
| Total Global RMSF | 91.8492 | 91.5704 | 91.8067 | 91.0109 |
| Average Global RMSF | 0.1812 | 0.1806 | 0.1811 | 0.1795 |
| Total Regional RMSF(Pocket 1) | 5.4497 | 5.3549 | 5.3004 | 5.4541 |
| Average Regional RMSF(Pocket 1) | 0.1758 | 0.1727 | 0.171 | 0.1759 |
| Highest Fluctuation | 0.3149 | 0.2809 | 0.309 | 0.2884 |
| Least Fluctuation | 0.1038 | 0.1244 | 0.1183 | 0.122 |
| Range of RMSF | 0.2111 | 0.1565 | 0.1907 | 0.1664 |
| **PCA (motions)** | | | | |
| Total global (mean of PC1, PC2 & PC3) | 20.3527 | 20.5002 | 20.5907 | 20.4168 |
| Average global (mean of PC1, PC2 & PC3) | 0.0401 | 0.0404 | 0.0406 | 0.0403 |
| Total Regional (mean of PC1, PC2 & PC3) | 1.3102 | 1.2451 | 1.1877 | 1.2211 |
| Average Regional (mean of PC1, PC2 & PC3) | 0.0423 | 0.0402 | 0.0383 | 0.0394 |
| Best global Conformation | PC3 | PC3 | PC2 | PC2 |
| Best regional Conformation (Pocket 1) | PC3 | PC3 | PC2 | PC2 |
| PC1 Eigenvalue | 6.74 | 6.76 | 6.59 | 6.7 |
| PC2 Eigenvalue | 6.43 | 6.36 | 6.36 | 6.43 |
| PC3 Eigenvalue | 6.31 | 6.12 | 6.11 | 6.15 |
| Total | 19.48 | 19.24 | 19.06 | 19.28 |
| **B-Factor** | | | | |
| Global average B factor | 12.2297 | 27.7477 | 10.4732 | 15.6061 |
| Regional average B factor | 5.611 | 11.234 | 5.6169 | 7.1471 |
| **Radius of Gyration** | | | | |
| Average Gyration | 5.6951 | 5.6966 | 5.6951 | 5.6941 |
| Minimum Gyration | 5.693 | 5.6943 | 5.6908 | 5.6915 |
| Maximum Gyration | 5.6965 | 5.6984 | 5.6998 | 5.6966 |
| Range Gyration | 0.0035 | 0.0041 | 0.009 | 0.0051 |
| % Gyration | 0.061 | 0.072 | 0.158 | 0.09 |
| Time Frame of Maximum Gyration | 19 | 15 | 7 | 14 |
| Time Frame of Minimum Gyration | 13 | 12 | 3 | 20 |
| **DCCM** | | | | |
| Anti-correlating residues | 16 | 19 | 17 | 17 |

**Table 7.  BLAST result for the homologues of the target protein in the human specie.**

| Accession | Name | Accession length | Max Score | Total Score | Query cover | E-value | % identity |
|---|---|---|---|---|---|---|---|
| CBX51367.1 | unnamed protein product | 900 | 148 | 148 | 91% | 4.00E-38 | 25.83% |
| CAE90564.1 | unnamed protein product | 567 | 47 | 47 | 22% | 1.00E-05 | 34.43% |
| CAE89160.1 | unnamed protein product | 764 | 45.1 | 45.1 | 22% | 5.00E-05 | 35.25% |

most soluble of the compounds. With water solubility values less than -4.0 log mol/L, OOU and Strophanthidin are poorly soluble [47]. Oral administration remains the primary method of drug administration, making in-vitro permeability studies useful for predicting oral bio-availability. The Caco-2 cell monolayers, which produce tight connections between cells, are employed as a model of human intestinal absorption because they closely resemble the human intestinal epithelium in many ways [64]. Isopteropodin showed the highest Caco-2 permeability. The OOU also has high Caco-2 permeability while that of Strophanthidin is low [47]. Similarly, the determination of human intestinal absorption (HIA) is a very important aspect in the creation of novel pharmacological compounds [65]. Though all the compounds have high percentage HIA, Isopteropodin has the highest value [47]. For effective transdermal delivery, it is necessary to assess drug penetration through the skin [66]. While all the compounds had skin permeability (LogKp) values less than 2.5, Strophanthidin has the best dermal permeability value.

The P-glycoprotein (Pgp) which has an influence on ADMET properties is a unidirectional efflux pump that removes its substrate such as drugs, pollutants, and other xenobiotics from inside to outside of the cells [67]. All the compounds are Pgp substrates and this implies that their oral bioavailabilities would be reduced by Pgp. Unlike the lead compounds, OOU is predicted to be Pgp I and II inhibitors suggesting that it would facilitate the intracellular accumulation of substrates leading to toxicity [68].

The volume of distribution steady state (VDSS) is an important pharmacokinetic property that determines the dosing frequency and half-life of a drug [69]. The VDSS for OOU is extremely high requiring about 8.41l/kg; the VDSS of Isopteropodin is high requiring about 5.28 l/kg; and the VDSS of Strophanthidin is low requiring about 0.89 l/kg to maintain uniform distribution to give the same concentration in the plasma [47]. The degree to which a drug binds to plasma proteins has an impact on its efficacy [47]. Though they all exceeded 0.1, the values of fraction unbound (human) for OOU suggests that it is the least available for bioactivity. Though Isopteropodin and Strophanthidin have similar values, Strophanthidin is more available [47]. The blood–brain barrier (BBB) prevents the uptake of most drugs. However, certain drugs with unique chemical properties are able to cross the BBB through lipid-mediated free diffusion [70]. From Table 2, all compounds have their BBB permeability (log BB) larger than -1.0 but less than 0.3 suggesting that they are all moderately distributed in the brain. The OOU is predicted to have the best brain distribution while Strophanthidin has the poorest [47]. Similarly, Strophanthidin is also unable to permeate the CNS, while Isopteropodin and OOU can moderately permeate it [47]. In-vivo study may be needed to determine if this moderate permeability may be effective in the treatment of neurobrucellosis which is a complication due to chronic brucellosis.

Cytochromes P450 (CYP) is responsible for the biotransformation of most drugs and is a primary cause of variability in drug pharmacokinetics. The CYPs 3A4, 2C9, and 1A2 are the most prevalent in the liver, while 2D6 and 2C19 are less abundant [71]. All compounds are substrates of CYP450 3A4 and only OOU is a substrate of CYP450 2D6. This suggests that these metabolic enzymes facilitate the biotransformation of these compounds making them

available for excretion. Remarkably, OOU is an inhibitor of CYP450, P450 2D6, 1A2, and 3A4 causing the accumulation of the substrate of these enzymes [47].

The total clearance of a drug from the bloodstream is the sum of the renal clearance, the hepatic clearance, and the clearance from all other tissues [72]. Depending on the functionality of the organs involved and several other factors, the total clearance ranges from 0 to 1.0. The results as indicated in Table 2 show that OOU followed by Isopteropodin have a very high rate of elimination from the plasma while that of Strophanthidin is slowest.

The renal organic cation transporter 2 (OCT2) protein is found in the basolateral membrane of proximal epithelial cells and it is involved in cationic drug uptake and secretion [73]. Only Strophanthidin, as shown in Table 2, will not be carried from the plasma into the cells of the proximal convoluted tubule by the renal OCT2 and will also have no deleterious interactions when co-administered with renal OCT2 inhibitors [47].

The potassium channel protein expressed by the human ether-a-go-go related gene (hERG) is important for cardiac repolarization and arrhythmias caused by long QT wave [74]. The study also found that only OOU is predicted to be an inhibitor of hERG II protein showing its potential cardiotoxic property [47]. However, all the compounds are neither genotoxic nor dermato-toxic.

As established by early-stage human clinical trials, the maximum tolerated dose (MTD) of a drug is the highest dose of that drug that does not induce overt toxicity or undesirable side effects within a set time frame [75]. In the present study, all the compounds have low MTD being lower than 0.477 log (mg/kg/day) [47]. The oral rat chronic toxicity is the lowest dose of a drug that results in an observed adverse effect over a time period, while the oral rat acute toxicity or $LD_{50}$ is the measurement of how much of a drug is required to kill 50% of rats in a test [47]. In terms of acute toxicity, Isopteropodin is the safest, while Strophanthidin is safest in terms of chronic toxicity. Similarly for toxicity to *Tetrahymena pyriformis*, Isopteropodin is the safest while for toxicity to Minnows, Strophanthidin is the safest. Despite the fact that the liver is the most common target organ for drug candidates in animal toxicity tests, hepatotoxicity seldom causes drug development to be halted during the preclinical stage. When a drug has great therapeutic promise, hepatotoxicity in humans may be tolerable due to the fact that it is frequently reversible and dose dependent [76]. In this study, only Strophanthidin is predicted to be non-hepatotoxic.

## Analyses of time-resolved trajectories

The RMSD calculates the differences in distances between atoms in two stacked protein structures (the reference and target) with a result of 0.0 indicating perfect overlap [77]. Over the 2-nanosecond trajectory, the BrMelMetRS-OOU and BrMelMetRS-Isopteropodin complexes showed marginally greater distortion than the BrMelMetRS-Strophanthidin complex in terms of variations in the RMSD of the Cα atomic coordinates. This is evidenced by the values of highest RMSD peak, the total RMSD, and the average RMSD. All the RMSD slopes induced by the holo forms show a gentle upward trend suggesting greater values with more simulation time. In this study, as it concerns RMSD peaks distribution patterns, all the holo forms show similar stability [78]. The structure and dynamics of proteins also play a big role in how well they work. The Root mean square fluctuation (RMSF) measures the structural flexibility of the protein by calculating the fluctuations of residues during molecular dynamics simulation [79,80]. While BrMelMetRS- Isopteropodin complex showed the greatest fluctuations amongst the holo structures at the global level, the BrMelMetRS-Strophanthidin complex showed the greatest fluctuations at the regional level (Pocket 1).

The PCA is used to statistically evaluate the various structural conformations of a protein generated during trajectories [81]. This study found that the BrucMetRS—Isopteropodin

complex has the greatest global (total and average) motions of any holo structures, closely followed by the BrMelMetRS-OOU complex. At Pocket 1, the BrMelMetRS-OOU complex showed the highest regional (total and average) motions whereas, the BrMelMetRS-Strophanthidin complex showed greater regional motions than the BrucMetRS-Isopteropodin complex. Specifically, based on the highest motions, the best global and regional conformations are PC3, PC2, and PC2 for the BrMelMetRS-OOU, BrucMetRS–Isopteropodin, and the BrMelMetRS-Strophanthidin complexes respectively.

The B-factor is a measurement of a protein's thermal stability based on the variation in atom locations in relation to average atomic coordinates [82]. Of all the holo structures, the BrMelMetRS-OOU complex showed the highest B-factor suggesting the greatest thermal instability. However, BrMelMetRS- Strophanthidin complex showed greater thermal instability than the BrucMetRS—Isopteropodin complex as seen by the global and regional average B factor values. The radius of gyration is the determinant of the compactness of the apo or holo protein during molecular dynamics simulation [83]. In terms of RoG along the trajectory, Isopteropodin induced the least compactness on the target (S4 Fig, Table 6)

The dynamic cross-correlation map depicts the atomic correlation pattern in protein dynamics [84]. Of the 31 residues of the Pocket 1, the BrMelMetRS-OOU complex showed the highest number of anti-correlating residues. The BrMelMetRS- Strophanthidin and the BrucMetRS—Isopteropodin complexes have the same number of anti-correlating residues. The net values for all the residues in the Pocket 1 reveal that the Strophanthidin had the greatest anti-correlation effect on the target protein suggesting the greatest inhibitory activity at that site [58].

## BLAST

Many drugs are quite promiscuous and they would bind to several targets with structural similarity [85]. Fortunately, the bacterial methionyl-tRNA synthetase (MetRS) enzyme, which is required for protein synthesis, differs significantly from the human cytoplasmic equivalent (HCE) and therefore the HCE would not be inhibited by the lead compounds [86]. However, there is a possibility that the lead compounds interact with the unnamed protein product, CBX51367.1 which though has less than 30% identity, but has an E value less than $10^{-6}$ sharing significant similarity with BrucMetRS [87].

Taken together, this study demonstrates the potential antibacterial effect of the reference compound OOU, and the leads compounds, Isopteropodin and Strophanthidin. However, OOU is slightly unsaturated therefore showing poor drug likeness and the ability to inhibit P-glycoprotein I and II proteins. Both Isopteropodin and Strophanthidin have shown acceptable pharmacokinetic properties with Isopteropodin showing superior oral absorbability. In terms of time-resolved trajectory of the apo and holo structures of the target, Strophanthidin induced the greatest molecular distortion at Pocket 1 as seen with the RMSF, PCA, B-factor and DCCM results.

Strophantidin is a cardiac glycoside found in the seed of edible plant, *Corchorus olitorius*, and has been used in the treatment of congestive heart failure. It functions by inhibiting the membrane bound Na+/ K+ ATPase in the cardiac muscles [88,89]. This blockage leads to influx of calcium ions leading to an inotropic effect. This mechanism of action is dose-dependent (0.1 μmol/L and 0.5 μmol/L), as Strophanthidin can be potentially cardiotoxic through Ca2+ overload, diastolic dysfunction, and arrhythmias when administered above maximum dose [88]. The anticancer potential of Strophanthidin has also been identified as it inhibits the MAPK, PI3K/AKT/mTOR, and Wnt/β-Catenin signaling Pathways [90]. Further experiments

are required to ascertain whether sub-therapeutic doses of Strophanthidin can induce significant antibacterial effect *in-vivo*.

Isopteropodin is an oxindole alkaloid isolated from the Cat's claw plant (*Uncaria tomentosa)* whose water-soluble extract significantly enhanced immune function by increasing Phytohemagglutinin (PHA) stimulated lymphocyte proliferation in splenocytes of rats [91,92] The findings of this study suggest that the plants, *Corchorus olitorius*, and *Uncaria tomentosa* containing the lead compounds, Strophantidin and Isopteropodin respectively could be exploited to make antibiotics for the treatment of brucellosis.

## Limitations of the study

The study did not test the effectiveness of the compounds in-vitro and in-vivo against *B. melitensis* and therefore, the $LD_{50}$ was not determined.

## Conclusion

This study indicates that Isopteropodin and Strophanthidin have the capacity to block the *Brucella mellitensis* Methionyl-tRNA synthetase at Pocket 1. Therefore, they could be possible drug candidates for the treatment of brucellosis and hence have a high potential for clinical development. This paves the way for subsequent in-vitro and in-vivo studies using animal models to determine the effectiveness and toxicity of the lead compounds.

## Supporting information

**S1 Fig. Root mean square deviations of the apo and holo forms of the target.**
(TIF)

**S2 Fig. RMSD histogram of the apo and holo forms of the target.** (a) BrMelMetRS (b) BrMelMetRS-OOU complex (c) BrMelMetRS-Isopteropodin complex (d) BrMelMetRS-Strophanthidin complex.
(TIF)

**S3 Fig. RMSF of the apo and holo forms of the target.** (a) BrMelMetRS (b) BrMelMetRS-OOU complex (c) BrMelMetRS-Isopteropodin complex (d) BrMelMetRS-Strophanthidin complex.
(TIF)

**S4 Fig. PCA: Cluster plots of the apo and holo forms of the target.** The trajectory projection onto the first three eigenvectors for: (a) BrMelMetRS (b) BrMelMetRS-OOU complex (c) BrMelMetRS-Isopteropodin complex (d) BrMelMetRS-Strophanthidin complex.
(TIF)

**S5 Fig. Radius of Gyration for the apo and holo forms of the target.** (a) BrMelMetRS (b) BrMelMetRS-OOU complex (c) BrMelMetRS-Isopteropodin complex (d) BrMelMetRS-Strophanthidin complex.
(TIF)

**S6 Fig. B-factor of the apo and holo forms of the target.** (a) BrMelMetRS (b) BrMelMetRS-OOU complex (c) BrMelMetRS-Isopteropodin complex (d) BrMelMetRS-Strophanthidin complex.
(TIF)

**S7 Fig. Dynamic cross correlation matrix of the apo and holo forms of the target.** Dark cyan represents fully correlated motion, purple represents anti-correlated motion, while white

and cyan represent moderately and uncorrelated motions respectively. Values of -1.0 are anti-correlated motion; 0 is non-correlated motion; and 1.0 is correlated motion. (a) BrMelMetRS (b) BrMelMetRS-OOU complex (c) BrMelMetRS-Isopteropodin complex (d) BrMelMetRS--Strophanthidin complex.
(TIF)

**S1 Table. Summary of the computational results.** Table A. The results of the molecular docking between target and library of natural compounds. Table B. Data on the chemical and physical properties of reference and lead compounds. Table C. The molar refractivity, saturation, and promiscuity profiles of front-runner compounds. Table D. The ADMET properties of front-runner compounds. Table E. The bioactivities of reference and lead compounds on different drug targets. Table F. The physicochemical properties of reference and lead compounds. Table G. The amino acids found in the binding pockets of the target protein. Table H. The RMSD data from the apo and holo forms of the target. Table I. The RMSF data from the apo and holo forms of the target. Table J. The PCA data from the apo and holo forms of the target. Table K. The DCCM data from the apo and holo forms of the target. Table L. The B-factor data from the apo and holo forms of the target. Table M. The radius of gyration data from the apo and holo forms of the target. Table N. Summary of data after MDS of the apo and holo forms of the target. Table O. BLAST result for the homologues of the target protein in the human species.
(XLSX)

## Author Contributions

**Conceptualization:** Adekunle Babajide Rowaiye, Akwoba Joseph Ogugua.

**Data curation:** Adekunle Babajide Rowaiye, Akwoba Joseph Ogugua, Gordon Ibeanu, Doofan Bur, Mercy Titilayo Asala, Osaretin Benjamin Ogbeide, Emmanuella Oshiorenimeh Abraham, Hamzah Bundu Usman.

**Investigation:** Gordon Ibeanu, Osaretin Benjamin Ogbeide.

**Methodology:** Adekunle Babajide Rowaiye, Gordon Ibeanu, Doofan Bur, Mercy Titilayo Asala, Osaretin Benjamin Ogbeide, Emmanuella Oshiorenimeh Abraham, Hamzah Bundu Usman.

**Project administration:** Akwoba Joseph Ogugua.

**Resources:** Adekunle Babajide Rowaiye, Akwoba Joseph Ogugua.

**Software:** Gordon Ibeanu.

**Supervision:** Adekunle Babajide Rowaiye, Akwoba Joseph Ogugua.

**Validation:** Adekunle Babajide Rowaiye, Akwoba Joseph Ogugua, Doofan Bur, Mercy Titilayo Asala, Emmanuella Oshiorenimeh Abraham, Hamzah Bundu Usman.

**Writing – original draft:** Adekunle Babajide Rowaiye, Akwoba Joseph Ogugua, Gordon Ibeanu, Osaretin Benjamin Ogbeide.

**Writing – review & editing:** Adekunle Babajide Rowaiye, Akwoba Joseph Ogugua, Gordon Ibeanu, Doofan Bur, Osaretin Benjamin Ogbeide, Hamzah Bundu Usman.

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
