## [Decision Letter · Decision Letter 0]

4 Jan 2022

Dear Dr. Ogugua,

Thank you very much for submitting your manuscript "Identifying potential natural inhibitors of Brucella melitensis Methionyl-tRNA synthetase through an in-silico approach" for consideration at PLOS Neglected Tropical Diseases. As with all papers reviewed by the journal, your manuscript was reviewed by members of the editorial board and by several independent reviewers. In light of the reviews (below this email), we would like to invite the resubmission of a significantly-revised version that takes into account the reviewers' comments. 

We cannot make any decision about publication until we have seen the revised manuscript and your response to the reviewers' comments. Your revised manuscript is also likely to be sent to reviewers for further evaluation.

Sincerely,

Tao Lin, DVM, MSc

Associate Editor

Godfred Menezes

Deputy Editor

Reviewer's Responses to Questions

**Key Review Criteria Required for Acceptance?**

**Methods**

-Are the objectives of the study clearly articulated with a clear testable hypothesis stated?

-Is the study design appropriate to address the stated objectives?

-Is the population clearly described and appropriate for the hypothesis being tested?

-Is the sample size sufficient to ensure adequate power to address the hypothesis being tested?

-Were correct statistical analysis used to support conclusions?

-Are there concerns about ethical or regulatory requirements being met?

Reviewer #1: Are the objectives of the study clearly articulated with a clear testable hypothesis stated? - Yes

-Is the study design appropriate to address the stated objectives? - Yes

-Is the population clearly described and appropriate for the hypothesis being tested? - Yes

-Is the sample size sufficient to ensure adequate power to address the hypothesis being tested?

-Were correct statistical analysis used to support conclusions? - Yes

-Are there concerns about ethical or regulatory requirements being met? - No

Reviewer #2: I have been asked to review the manuscript entitled, "Identifying potential natural inhibitors of Brucella melitensis Methionyl-tRNA synthetase through an in-silico approach". 

The authors used an in-silico approach to identify chemical compounds that originate in plants that would bind and thus be inhibitors of specific tRNA synthatases in B. melitensis. This is not a novel approach per se, yet it is for Brucella spp research. The over 1,500 phytoproteins were identified and then screened using specific software using the compatibility of both the enzyme and the substrates. 

The aim of the study is laid out in the introduction section and is well well described. However, the hypothesis per se is not defined in the body of the introduction. The description as to why this study was undertaken, but as to what the authors speculated would happen in their research was not delineated.

Another issue for the authors to address was what was the rationale for the >1,500 proteins being analyzed from African plants. Was there a hypothesis associated as to why the authors decided this? If so, this was not documented. It would be intriguing if the authors had some information to explain this decision.

**Results**

-Does the analysis presented match the analysis plan?

-Are the results clearly and completely presented?

-Are the figures (Tables, Images) of sufficient quality for clarity?

Reviewer #1: Does the analysis presented match the analysis plan? - Yes

-Are the results clearly and completely presented? - Yes

-Are the figures (Tables, Images) of sufficient quality for clarity? - Yes

Reviewer #2: The stereochemical analysis done by the authors is illustrated well through the use of the figures and tables. There are some issues with the figures themselves where the figure legends need to be much more explanatory for such a study. A good example of this is Figure 3, where the description in the figure legend is brief and is not descriptive of the color scheme used as well as a formal chemical formula being presented as well (even though the latter is in Table 1). 

Many of the figures could be in supplemental figure sets. This is especially true for Figures 8-14. In addition, the figure legends of Figures 8-14 need more explanation than what is provided. 

The description of the compounds, including their potential affect on P450 system, and the volume distribution and hypothesized toxicities were all well explained. Each are described briefly when it comes to their toxicities; there are some potential issues with these compounds and even though they might be effective inhibitors of BrMelMetRS, they could possibly present some challenges with the toxicities.

**Conclusions**

-Are the conclusions supported by the data presented?

-Are the limitations of analysis clearly described?

-Do the authors discuss how these data can be helpful to advance our understanding of the topic under study?

-Is public health relevance addressed?

Reviewer #1: Are the conclusions supported by the data presented? - To a large extent

-Are the limitations of analysis clearly described? - None

-Do the authors discuss how these data can be helpful to advance our understanding of the topic under study? - Yes

-Is public health relevance addressed? - Yes

Reviewer #2: The Discussion section is well written and lays out the three target effector molecules with vernacular which would be relevant to all of the readers of PLOS NTD. There are NO limitations per se in the manuscript and even though there is discussion regarding the limitations of the three compounds, there needs to be more description of what could gave gone wrong in their analysis or what their next steps would be. 

The need for a public health relevance is discussed in the Introduction, but there is no translational science description here in the Discussion/Conclusion sections. More needs to be documented in this area, for this to provide more of an understanding for the readers of PLOS NTD

**Editorial and Data Presentation Modifications?**

Reviewer #1: (No Response)

Reviewer #2: Besides what is described above when it comes to the figures, and the figure legends, the paper is well written with appropriate vernacular. I do think that Figures 8-14 should be supplementary in nature.

**Summary and General Comments**

Reviewer #1: Introduction

There is need to provide more literature on the potentials or use of medicinal plants in the control of infectious diseases, as a basis for the present investigation. Are the preset drugs of choice for brucellosis readily available and accessible? What is the magnitude of the drug resistance challenge?

Methods

How was safety of the compounds determined? Any LD50?

No information on ethical approval for the study

Results

Line 169: The first table cited is referenced “Table 4”, whereas, no table was previously cited.

Discussion

The discussion is filled with more of literature review than discussing the findings. I suggest reducing some of the literature review in this section. I recommend reducing the length of discussion section and make it more precise and succinct.

Conclusion

Concluding that the compounds could be used to treat human brucellosis is too ambitious. Exploring the compounds in animal models is required to validate this statement.

Reviewer #2: After reviewing "Identifying potential natural inhibitors of Brucella melitensis Methionyl-tRNA synthetase through an in-silico approach", I believe that this will be a publication worth having the readers of PLOS NTD have the opportunity to read. The strengths of this manuscript are the novelty of the study (as it relates to Brucella melitensis, even though this approach itself is not novel) and the significance since it is the most common zoonosis on globe. The study done by the authors demonstrated that they understand the stereochemistry and how it should be applied through a specific target, which in this case is a tRNA synthetase of B. melitensis. Based on the analysis, they identified three substrate molecules which have the best likelihood of inhibition of this tRNA synthetase, as well as why each one would have an advantage/disadvantage over the other two. 

The issues with the study are not really related to execution but to details within the manuscript. First, the introduction/methods does not really elucidate as to why only 1,500 phytochemicals from African plants were selected, as opposed to other compounds that are either organic/inorganic. The rationale here is critical for the readers to know that the results aren't biased and that there could be other compounds out there that are more affective in inhibiting the tRNA synthetase. 

The other issues are related to the data regarding the stereochemistry and their application to the manuscript in toto. These should be referenced and placed in supplemental materials. Also, the figure legends need to be more descriptive in their nature. Furthermore, there are no limitations noted in the manuscript and there are no future steps to be undertaken by this group. This is critical to know that the authors are trying to push the science forward and looking towards their next project. Finally, this is reflective in the conclusions as well. This should be much more descriptive in why this study was important and what they want to do next.

PLOS authors have the option to publish the peer review history of their article (what does this mean?). If published, this will include your full peer review and any attached files.

Reviewer #1: No

Reviewer #2: No
---

## [Editor Report · Decision Letter 1]

16 Feb 2022

Dear Dr. Ogugua,

We are pleased to inform you that your manuscript 'Identifying potential natural inhibitors of Brucella melitensis Methionyl-tRNA synthetase through an in-silico approach' has been provisionally accepted for publication in PLOS Neglected Tropical Diseases.

Best regards,

Tao Lin, DVM, MSc

Associate Editor

Godfred Menezes

Deputy Editor

---

## [Editor Report · Acceptance letter]

8 Mar 2022

Dear Dr. Ogugua,

We are delighted to inform you that your manuscript, "Identifying potential natural inhibitors of Brucella melitensis Methionyl-tRNA synthetase through an in-silico approach," has been formally accepted for publication in PLOS Neglected Tropical Diseases.

Best regards,

Shaden Kamhawi

co-Editor-in-Chief

Paul Brindley

co-Editor-in-Chief
